# UV-DDB as a General Sensor of DNA Damage in Chromatin: Multifaceted Approaches to Assess Its Direct Role in Base Excision Repair

**DOI:** 10.3390/ijms241210168

**Published:** 2023-06-15

**Authors:** Sripriya J. Raja, Bennett Van Houten

**Affiliations:** 1Molecular Pharmacology Graduate Program, School of Medicine, University of Pittsburgh, Pittsburgh, PA 15213, USA; srr66@pitt.edu; 2UPMC Hillman Cancer Center, University of Pittsburgh, Pittsburgh, PA 15213, USA; 3Department of Pharmacology and Chemical Biology, School of Medicine, University of Pittsburgh, Pittsburgh, PA 15213, USA

**Keywords:** DNA damage, base excision repair, nucleotide excision repair, UV-DDB, chromatin, nucleosome, DNA glycosylases, single molecule, cell biology, biochemistry

## Abstract

Base excision repair (BER) is a cellular process that removes damaged bases arising from exogenous and endogenous sources including reactive oxygen species, alkylation agents, and ionizing radiation. BER is mediated by the actions of multiple proteins which work in a highly concerted manner to resolve DNA damage efficiently to prevent toxic repair intermediates. During the initiation of BER, the damaged base is removed by one of 11 mammalian DNA glycosylases, resulting in abasic sites. Many DNA glycosylases are product-inhibited by binding to the abasic site more avidly than the damaged base. Traditionally, apurinic/apyrimidinic endonuclease 1, APE1, was believed to help turn over the glycosylases to undergo multiple rounds of damaged base removal. However, in a series of papers from our laboratory, we have demonstrated that UV-damaged DNA binding protein (UV-DDB) stimulates the glycosylase activities of human 8-oxoguanine glycosylase (OGG1), MUTY DNA glycosylase (MUTYH), alkyladenine glycosylase/N-methylpurine DNA glycosylase (AAG/MPG), and single-strand selective monofunctional glycosylase (SMUG1), between three- and five-fold. Moreover, we have shown that UV-DDB can assist chromatin decompaction, facilitating access of OGG1 to 8-oxoguanine damage in telomeres. This review summarizes the biochemistry, single-molecule, and cell biology approaches that our group used to directly demonstrate the essential role of UV-DDB in BER.

## 1. Introduction

Base damage to DNA is caused by endogenous and exogenous sources and if left unrepaired can lead to a plethora of diseases including cancer, cardiovascular disease, neurodegeneration, and premature aging [1,2,3]. The human genome is approximately three billion base pairs in size and is constantly subjected to assault by these damaging agents. Single base damage is processed by the base excision repair (BER) pathway, which has been fully reconstituted with purified proteins on DNA templates containing site-specific damage [4,5]. In general, BER begins when the lesion-specific glycosylase cleaves the glycosidic bond, creating an abasic (AP) site. The AP site is processed by apurinic/apyrimidinic endonuclease 1(APE1), creating a single-strand break that is bound by PARP1. PARP1 is then activated and begins building poly-ADP-ribose (PAR) chains to signal downstream recruitment of later BER proteins, including DNA polymerase β (Pol β) and ligase 3 (LIG3), to fill in the correct base and seal the resulting nick [1,6].

The human genome is compacted into chromatin within the cell nucleus, with the central repeating unit of chromatin being the nucleosome, in which 147 base pairs of DNA are wrapped ~1.67 times around an octamer of histones (H2A, H2B, H3, and H4). The center point of the nucleosome is defined as the dyad axis [7], and each turn of the helix is noted as a superhelical location (SHL), with the dyad being SHL0 and positions being 1 to 7 in one direction and −1 to −7 in the other direction. While nucleosome packaging can protect the DNA from oxidative and radiation damage [8], the position of DNA within this structure poses an obstacle to BER. Previous work by several groups including the Delaney, Hinz, Menoni, Smerdon, Wallace, and Wilson laboratories, demonstrated that many BER enzymes are inhibited when a DNA lesion is embedded in a nucleosome [9,10,11,12,13,14,15,16,17]. Early work from the Wilson and Smerdon groups looked at the enzymatic activities of human uracil DNA glycosylase (UDG), APE1, and Pol β on reconstituted nucleosomes created from satellite DNA containing uracil in different rotational positions (inward and outward) [9]. In this study, they demonstrated that UDG and APE1 work less efficiently on uracil embedded within the nucleosome, as compared with naked DNA. Importantly, UDG and APE1 had over a two-fold increase in enzymatic activities on the uracil in the outward position compared with that in the inward position. Interestingly, they also saw total inhibition of Pol β activity on nucleosomes, irrelevant to the rotational positioning of the lesion [9]. The Delaney group has demonstrated that base damage embedded in a 601 nucleosome is poorly recognized and processed by glycosylases [16,17,18]. The enzymatic activities of UDG, alkyladenine glycosylase (AAG), oxoguanine glycosylase 1 (OGG1), and bacterial proteins formamidopyrimidine [fapy]-DNA glycosylase (Fpg) and Endo III were assessed on reconstituted nucleosomes containing the targeted lesion of the repair enzyme [16]. They demonstrated that UDG and AAG were able to completely excise their respective lesions, U/5-OHU (uracil/5-hydroxyuracil) and Hx/εA (hypoxanthine/ethenoadenine), with high efficiency when the lesion was outward-facing on the nucleosome [16]. Interestingly, OGG1, FPG, and Endo III were only able to produce 10% of the excised product when their target lesions were embedded within nucleosomes. Using reconstituted nucleosomes containing 8-oxoguanine (8-oxoG) lesions, the Delaney group further assessed OGG1 activity and found that the greatest OGG1 activity occurred on lesions facing outward from the nucleosome, and conversely, the lowest activity was on the inward-facing lesions [18]. Moreover, they found that the positioning of the lesions on the dyad axis significantly reduced OGG1 activity regardless of the rotational positioning of the lesion [18]. As previously mentioned, UDG displayed high enzymatic activity on outward-facing U and 5-OHU lesions in nucleosomes. The Delaney group then measured the enzymatic activities of the other uracil DNA glycosylases, single-strand selective monofunctional DNA glycosylase (SMUG1) and thymine DNA glycosylase (TDG), on U:G mis-pairs embedded in nucleosomes [17]. In this study, it was demonstrated that UDG has the greatest activity on the U:G substrate in the nucleosome, while SMUG1 activity was reduced by 50% on this substrate when embedded in a nucleosome. Interestingly TDG exhibited enzymatic activity in between that of UDG and SMUG on the U:G substrate in the nucleosome. However, the activity of TDG on the U:G substrate was reduced by a factor of 50% when either a N-and-C-termini truncated TDG was used or when TDG was in the presence of a T:G mis-pair substrate on the nucleosome.

In general, the BER repair proteins can better process lesions that are facing out and away from the histone octamer core [10,12,14]. Specifically, it was demonstrated by the Hinz group that APE1 activity is diminished from 85% to 15% when abasic sites or abasic site analogs are incorporated in nucleosomes containing the 601 sequence or the TG positioning sequence [10]. Lesions that are located more inward towards the octamer core can be accessed by the dynamic unwrapping of the nucleosome. Lastly, when the lesion is located close to the dyad axis it is repaired less efficiently [6,19]. Since BER enzyme activity is reduced when damage is located within a nucleosome, other factors within the cell may be responsible for increasing lesion access within chromatin. However, our current knowledge of how BER enzymes gain access to base damage in the context of chromatin in a living cell is less understood, and several studies have suggested the need for chromatin remodelers in an “access-repair-restore” model first proposed by Almouzni [20] and later supported by work from Thoma and Smerdon [21]. In the access step, chromatin remodelers work to reposition and reorganize nucleosomes to generate gaps to make the damage sites more accessible to repair factors. Once the damaged DNA is accessible, the damage is processed by the repair proteins, where the damaged base is excised, and a new nucleotide is inserted. Once the repair is complete, the chromatin remodelers restore the chromatin structure by reassembling the nucleosomes to maintain chromatin integrity. The finding of a reduction in glycosylase activity at the dyad axis supports the necessity of chromatin remodelers to increase lesion accessibility for repair factors, as previously demonstrated by Menoni and colleagues [15]. Menoni et al. studied 8-oxoG repair on conventional and histone H2A.Bbd variant nucleosomes containing the 601 sequence and demonstrated the need for SWI/SNF-mediated chromatin remodeling of conventional nucleosomes to promote 8-oxoG processing [15]. In this study, they first looked at the activities of OGG1 and APE1 on these reconstituted nucleosomes and demonstrated a 70–85% reduction in OGG1 and APE1 activities on nucleosomes. Moreover, they demonstrated that the addition of SWI/SNF resulted in a significant increase in APE1 and OGG1 activities. Lastly, this study looked at dGTP incorporation by Pol β as a measure of downstream 8-oxoG repair and found that SWI/SNF treatment was required for Pol β activity on the reconstituted nucleosomes [15].

Chromatin can be sub-characterized as euchromatin, transcriptionally active, or heterochromatin, transcriptionally repressed, to describe the level of accessibility and activity of the DNA [7]. It has been shown that chromatin remodelers and histone modifiers, including FACT, RSC, and ISWI, have roles in progressing BER [11,22]. Charles and Shukla et al. studied the role of FACT in BER in both purified and cellular systems [23]. Using a 405 nm laser and the photosensitizer Ro-19-8022 to generate localized 8-oxoG in Hela cells, they observed robust recruitment of OGG1 and SSRP1, a subunit of the dimeric FACT protein, after damage. Next, they treated Hela cells that were stably overexpressing HA-tagged SSRP1 with hydrogen peroxide and isolated chromatin-bound SSRP1 complexes, which were further digested with micrococcal nuclease to yield single SSRP1–nucleosome complexes. The complexes were then analyzed through Western blotting and mass spectrometry. Interestingly the mass spectrometry studies demonstrated that several DNA repair proteins could be pulled down along with SSRP1–nucleosome complexes after oxidative damage, notably DDB1, DDB2, CUL4A, SMARCA4, LIG3, XRCC1, and PARP1. Importantly, these proteins do not co-purify with SSRP1 in the absence of hydrogen peroxide treatment. They then examined the role of FACT in helping to initiate UDG-mediated BER on reconstituted nucleosomes. Interestingly, when they incubated purified UDG, APE1, and FACT together on the nucleosomal DNA, they saw no difference in the amount of product formation, suggesting that in this in vitro system FACT is not sufficient to provide full access to damage within a nucleosome. However, when UDG, APE1, FACT, and the chromatin remodeler RSC were incubated with the nucleosomal DNA, they observed an increase in lesion processing, returning to levels similar to that of naked DNA. Thus, their study supports a model in which FACT and RSC work in tandem to allow BER on damage embedded in nucleosomes.

In addition to lesion accessibility, the product of glycosylase activity, the AP site, can impede efficient repair since several glycosylases, such as OGG1, MUTYH, and SMUG1, have been shown to bind more tightly to the abasic site than to their substrate [24,25,26]. While APE1 is generally believed to provide both the next step in BER and promote efficient glycosylase turnover, not all glycosylases, such as MUTYH, can be displaced by the action of APE1 [27]. To help address the question as to whether other proteins beyond chromatin remodelers can help access base damage in chromatin and to determine whether other proteins can aid in glycosylase turnover from abasic sites, our laboratory has been focused on delineating the role of the heterodimeric protein UV-damaged DNA binding protein (UV-DDB) in the BER pathway [28,29,30,31,32]. Conventionally, UV-DDB (DDB1 and DDB2 subunits) works as the first responder during nucleotide excision repair (NER) to repair 6-4 photoproducts (6-4PP) and cyclobutane pyrimidine dimers (CPD) [33]. Upon UV damage, UV-DDB forms an E3 ubiquitin ligase complex with Culin4A and RBX to ubiquitylate histone H2A to destabilize the nucleosome and increase lesion accessibility for downstream NER proteins [34,35,36]. In the context of chromatin, it has been demonstrated that DDB2, the DNA binding subunit of UV-DDB, can promote the unfolding of heterochromatin by stimulating the displacement of linker histones from UV-damaged chromatin [37]. Polo and coworkers have shown that DDB2 is necessary and sufficient to decompact condensed chromatin associated with an LacI array [38].

Another unresolved question is the precise molecular role of PARP1 as a chromatin remodeler during NER at UV-damage sites [39,40]. After DNA damage, PARP1 activity, specifically PARylation of histones, has been shown to relax chromatin by facilitating the recruitment of chromatin remodelers to increase lesion accessibility for repair factors [41]. Recently, it was demonstrated that co-factor histone poly(ADP-ribosylation) factor 1 (HPF1) works with PARP1 to regulate and promote PARylation at histones after DNA double-strand breaks (DSB) to stimulate changes in chromatin structure to improve lesion accessibility during DNA DSB repair [42]. It is interesting to speculate that perhaps PARP1 has a similar role earlier in BER. There has been growing evidence of the interplay between BER and NER proteins during the repair of oxidative damage [43,44,45,46,47]. It has been demonstrated by several groups [48,49], including ours, that UV-DDB has a high affinity (Kd of ~1 nM) for AP sites [28]. This finding led our group to fully investigate the DNA damage repertoire of UV-DDB and whether this protein can act as a damage sensor in both NER and BER. This review summarizes recent work from our laboratory in establishing the direct role of UV-DDB in BER mediated by the glycosylases OGG1, MUTYH, AAG/MPG, and SMUG1, through biochemistry, single-molecule, and cell biology approaches [28,29,30,31,32].

## 2. Role of UV-DDB in 8-Oxoguanine Repair

### 2.1. Biochemistry Approach

Guanine has a high redox potential, and when exposed to reactive oxygen species, it leads to the lesion 8-oxoguanine (8-oxoG). The base excision repair protein, 8-oxoguanine glycosylase 1 (OGG1), works to remove 8-oxoG from the DNA to prevent G:C to T:A transversions. When adenine is mis-incorporated opposite an unrepaired 8:oxoG, the MUTYH removes the A, which is further processed by the long-patch BER pathway [50]. MUTY DNA glycosylase (MUTYH) is a monofunctional glycosylase, with no AP lyase activity, and has been shown to have a high affinity for the resulting abasic site [27]. While OGG1 is a bi-functional glycosylase, meaning it has both glycosylase and AP lyase activity, the AP lyase activity is weak. OGG1 has an inherent affinity for the abasic site product, causing it to become product inhibited. Both OGG1 and MUTYH require other proteins, such as APE1 or potentially UV-DDB, to promote glycosylase turnover. As previously mentioned, several groups have begun to delineate the interplay between NER and BER, as reviewed in [47]. Our group began by taking a biochemical approach to understand the range of DNA lesions that are recognized by UV-DDB. Sunbok Jang, a research associate in our laboratory, began by conducting electrophoretic mobility shift assays (EMSAs) with UV-DDB in both the presence and absence of magnesium (Mg^2+^) to determine the equilibrium dissociation constants (K_d_), a measure of binding affinity, for a variety of DNA damage substrates: CPD, a stable abasic site analog tetrahydrofuran (THF), 8-oxoG:C, and 8-oxoG:A. (Figure 1).

The EMSA experiments revealed that UV-DDB, in the presence of Mg^2+^ (5 mM), bound avidly to CPD, THF, and 8-oxoG mis-pairs with equilibrium dissociation constants, K_d_, of 3.9, 30, and 160 nM, respectively, with much greater affinity as compared with undamaged DNA (K_d_ = 1108 nM). Jang then wanted to know if UV-DDB could promote the enzymatic activities of OGG1, APE1, and MUTYH [28,30]. To address this question he performed excision assays with 8-oxoG:C, THF, and 8-oxoG:A DNA substrates for OGG1, APE1, and MUTYH, respectively, in both the presence and absence of purified UV-DDB. Remarkably, he was able to observe three-fold, eight-fold, and seven-fold stimulation of OGG1, APE1, and MUTYH, respectively, by UV-DDB (Figure 2) [28,30].

### 2.2. Single Molecule Approach

Our laboratory has used three different single-molecule approaches to understand the mechanism of UV-DDB stimulation of these BER enzymes. The first experimental approach used the DNA tightrope assay to assess the dynamics of OGG1, APE1, and MUTYH with UV-DDB on THF DNA. In the DNA tightrope assay, DNA containing the abasic site analog (THF) every 2 kb was strung up between poly-L-lysine beads. The proteins of interest, OGG1, APE1, and MUTYH, were labeled with a green 605-Qdot, and UV-DDB was labeled with a red 705-Qdot and flowed onto the DNA substrate. Using this approach, Jang was able to determine the binding lifetimes and co-localization events in real time of OGG1, APE1, and MUTYH with UV-DDB on abasic sites on DNA. Moreover, he was able to show that increasing concentrations of UV-DDB promote the release of OGG1, APE1, and MUTYH from the abasic site using a facilitated dissociation mechanism. While these experiments suggested the formation of a co-complex between MUTYH and UV-DDB, we were unable to distinguish the direct interaction between these proteins using size-exclusion chromatography [30]. To confirm co-complex formation between UV-DDB and MUTYH at sites of 8-oxoG: A damage, we turned to another single-molecule approach, atomic force microscopy (AFM). Using AFM, a Ph.D. student in the laboratory, Brittani Schnable, was able to study the binding of UV-DDB and MUTYH on a 538 bp duplex DNA containing an 8-oxoG:A pair 30% from one end of the duplex. By measuring the volumes of the UV-DDB and MUTYH complexes and correlating the measurements to a standard curve with known molecular weights, she was able to conclude that UV-DDB and MUTYH interact at 8-oxoG:A lesion sites [30]. The last single-molecule approach we utilized to demonstrate the involvement of UV-DDB in BER was the single-molecule analysis of DNA binding proteins from nuclear extracts (SMADNE) [51]. SMADNE, which was developed and validated by a postdoctoral fellow in the laboratory, Matthew Schaich, is an innovative method that allows for the delineation of DNA–protein interactions on the single-molecule level using fluorescently labeled proteins and optical tweezers combined with confocal microscopy. The LUMICKS C-trap used optical tweezers, microfluidics, and three-color confocal microscopy to allow us to determine event duration, event rate, the binding specificity, and velocity of DNA binding proteins, and thus was pivotal for the SMADNE approach. Using the C-trap, he was able to visualize OGG1-mScarlet interacting with dual-labeled UV-DDB (eGFP-DDB1 and HaloTag-JF635-DDB2), as well as the catalytically dead OGG1 mutant, K249Q, on DNA containing 8-oxoG (Figure 3) [51].

### 2.3. Cell Biology Approach

Next, our laboratory sought to define a direct role for UV-DDB in 8-oxoG repair in cells. A Ph.D. student, Namrata Kumar, used a chemoptogenetic system to generate 8-oxoG lesions specifically at the telomeres to determine if DDB2 is recruited to 8-oxoG in living cells. In this system, a fluorogen-activating peptide (FAP) was fused to telomeric repeat-binding factor 1 (TRF1), and in the presence of an MG2I dye and far-red 660 nm light, the FAP generated 8-oxoG at the telomeres through targeted singlet oxygen generation [52]. Utilizing the FAP system, we demonstrated robust recruitment of DDB2 and OGG1 at the telomeres after 8-oxoG damage via immunofluorescence (IF) staining in two different cell types [28,31]. Moreover, we were able to confirm the direct interaction between OGG1 and DDB2 at the telomeres after dye and light damage through proximity ligation assays (PLA) (Figure 4) [31]. We then wanted to better understand the dynamics of the interaction between DDB2 and OGG1. Through PLA, we demonstrated transient interactions between DDB2 and OGG1 after 8-oxoG damage that peaked 30 min post damage and were no longer detected after three hours post damage. These cellular interactions support our single-molecule studies showing that UV-DDB and OGG1 interact on DNA containing 8-oxoG or abasic sites [28,51]. To further strengthen our hypothesis that DDB2 directly interacts with the 8-oxoG damage site, we looked at the recruitment of the xeroderma pigmentosum E (XP-E) disease-causing DDB2 variant, K244E, which can no longer engage UV-induced photoproducts [53]. We observed a two-fold decrease in DDB2 recruitment to damaged telomeres. This reduction in binding was consistent with the inability of the variant to bind to UV-damage, emphasizing lysine 244 as an important residue for stabilizing DDB2 at sites of DNA damage (Figure 1). Using transient knockdown of DDB2, we observed a three-fold reduction in OGG1 recruitment after damage, providing clear evidence for the direct role of DDB2 as a damage sensor. We also observed that the knockdown of OGG1 resulted in a three-fold increase in DDB2 half-life at damage sites, indicating continued binding to 8-oxoG damage in the absence of OGG1 processing. These data collectively demonstrate that DDB2 is necessary for proper recruitment of OGG1 to 8-oxoG damage at the telomeres.

Nucleotide excision repair (NER) can be sub-characterized into two pathways, global-genome NER (GG-NER) and transcription-coupled NER (TC-NER) [34,36]. The pathways are distinguished by the proteins that recognize and initiate the repair. In GG-NER, the two proteins are UV-DDB and XPC-RAD23B-CEN2, whereas TC-NER begins when a stalled RNA polymerase II is recognized by CSA, CSB, and UV-SSA. The two pathways converge during the damage verification step facilitated by the multi-protein transcription factor, TFIIH. The TFIIH complex is stabilized at the damage site by several proteins, one being XPA [54]. Kumar next wanted to know if DDB2 facilitates the recruitment of XPC and XPA to sites of 8-oxoG damage, as previous works demonstrated roles for these NER proteins in the BER removal of 8-oxoG. Using IF, she saw peak recruitment of XPC and XPA 30 min post damage when DDB2 was present. Interestingly, the loss of DDB2 reduced XPC recruitment three-fold, but did not affect XPA recruitment after damage, demonstrating that XPC recruitment to lesion sites is dependent upon DDB2. As previously mentioned, chromatin dynamically regulates DNA transcription, with transcription occurring in euchromatic regions of the genome. While 8-oxoG is not a lesion that inhibits transcription elongation, the repair intermediates (AP sites or single-strand breaks) are strong blocks to transcription [55,56,57,58]. Previous work from the Spivak group demonstrated roles for XPA and TC-NER proteins in 8-oxoG repair at nicked DNA sites [59]. Kumar next sought to understand if XPC and XPA recruitment to 8-oxoG damage sites is dependent upon transcription. The transcription inhibitors, α-amanitin and THZ1, inhibited XPA recruitment after 8-oxoG damage more than two-fold, while XPC recruitment was unchanged after treatment with the transcription inhibitors. Furthermore, the loss of OGG1 reduced XPA accumulation by five-fold, consistent with XPA playing a direct role in some form of transcription-coupled BER in which RNA Pol II is stalled only after OGG1 and APE1 produces strand breaks.

To examine the role of DDB2 in the processing of 8-oxoG across the genome, we used the photosensitizer, Ro 19-8022, and a 405 nm laser pulse, in collaboration with a group at Erasmus University. This approach generated high densities of 8-oxoG at discrete foci and allowed real-time measurements of DDB2, OGG1, and XPC recruitment via live-cell imaging. Within a minute of 8-oxoG damage, they observed the recruitment of DDB2, OGG1, and XPC, which was recapitulated in three independent cell lines. They then modulated the active state of the Cullin4A-RBX-DDB2 E3 ubiquitin ligase complex through treatment with a neddylation (NEDD8) inhibitor (NAEi) or a COP9 signalosome inhibitor (CSN5i). Inhibition of neddylation kept the E3 ubiquitin ligase complex in an inactive state, while CSN5i treatment activated the E3 ubiquitin ligase complex and promoted DDB2 degradation. They were able to show that treatment with either NAEi or CSN5i reduced OGG1 recruitment after damage, showing that for efficient recruitment of OGG1 to sites of 8-oxoG throughout the genome, DDB2 needs to be active and present.

As previously mentioned, in the context of UV damage, UV-DDB forms an E3 ubiquitin ligase complex with Cullin4A and RBX to relax chromatin and increase lesion accessibility. However, the role of the E3 ligase complex in 8-oxoG repair was not yet characterized. Kumar first looked at DDB2 recruitment via IF and PLA after transient knockdown of Cullin4A or DDB1 and still observed DDB2 foci after damage. Furthermore, in WT cells, we observed less than 3% co-localization between DDB2 and either DDB1 or Cullin4A after damage. We then examined co-localization between DDB2 and Cullin4A in the absence of OGG1 and saw a two-fold increase in percent co-localization between the proteins. If DDB2 remains at a UV-damage site for an extended time, it triggers the recruitment of the E3 ubiquitin ligase to promote degradation via polyubiquitination [60]. As previously mentioned, in the absence of OGG1, DDB2 remains longer at the damage site, and this lingering of UV-DDB on DNA promotes Cullin4A recruitment to DDB2. Taken together, we showed that when OGG1 is present, DDB2 does not need the E3 ubiquitin ligase complex to actively engage with 8-oxoG lesions at the telomeres. DDB2 has known roles in chromatin decompaction after UV damage [37,38]. To elucidate if DDB2 can facilitate chromatin opening at damaged telomeres during 8-oxoG repair, we measured 3D volumes of telomeres in the presence of DDB2 following induction of 8-oxoG with confocal microscopy [31]. The volume measurements revealed an increase in telomere size in the WT cells with DDB2 present, which was not observed in the DDB2 KO cell lines. We demonstrated this change in telomere size after damage in two independent cell lines with DDB2 present. These experiments provide direct evidence that, in addition to helping glycosylases undergo multiple cycles of activity, DDB2 also helps relax chromatin to increase lesion accessibility for efficient repair of oxidative DNA damage [31].

## 3. Role of UV-DDB in Alkylation Damage N6-Ethenoadenine and Hypoxanthine Repair

Single base damage can arise from many different processes. Adenine is prone to alkylation damage, which can result in the production of N6-ethenoadenine (εA). Lipid peroxidation from chronic inflammation produces aldehydes that can react with DNA to damage adenine, generating the polymerase stalling lesion εA [61]. The deamination of adenine, which can also occur during pro-inflammatory responses such as chronic colitis, produces the mutagenic hypoxanthine (Hx), which if left unrepaired can lead to A:T to G:C transitions in the DNA [62]. These two base damages are repaired through the BER pathway initiated by the glycosylase Alkyladenine glycosylase (AAG), or N-methylpurine DNA glycosylase (MPG). As mentioned previously, glycosylases cleave the glycosidic bond between the DNA and the damaged base to create an abasic site product. Similar to OGG1, AAG has an affinity for AP sites, thus decreasing its overall catalytic efficiency [63]. Therefore, we wanted to understand if UV-DDB can promote the BER of alkylation damage by stimulating AAG activity [32]. Sunbok Jang in our group first used biochemistry experiments to show that UV-DDB can bind avidly to εA:T and Hx:T substrates. Using EMSAs, he determined the binding affinities of UV-DDB to εA:T and Hx:T substrates in the presence of 1.5 mM Mg^2+^ to be 6.0 nM and 8.6 nM, respectively, five- to six-fold better than undamaged DNA, which was determined to be 31.8 nM (Figure 1). After confirming that UV-DDB can bind to AAG substates, Jang then turned to excision assays to determine if UV-DDB can stimulate the enzymatic activities of AAG during BER [32]. He incubated AAG with either εA:T and Hx:T substrates in both the presence and absence of UV-DDB and observed a four- to five-fold increase in excision in the presence of UV-DDB (Figure 2). While previous studies supported the view that AAG is turned over by APE1, we found that UV-DDB increased AAG activity several fold, even when APE1 was present. As mentioned in the previous section, UV-DDB binds tightly to AP sites, which facilitates OGG1 dissociation from AP sites promoting BER. Using polyacrylamide gel electrophoresis (PAGE) assays, we incubated AAG (WT and a catalytically dead variant) with a THF (abasic site analog) substrate and added increasing amounts of UV-DDB, demonstrating complex formation between UV-DDB and AAG [32]. Furthermore, we were able to show UV-DDB freeing AAG from abasic sites, thus preventing re-binding and promoting repair. We then wanted to visualize this dynamic interaction between UV-DDB and AAG on the single-molecule level. Using the DNA tightrope assay, we observed co-localization between UV-DDB and AAG on THF-containing DNA. Additionally, we saw that UV-DDB decreased the half-life of AAG on abasic sites over eight-fold (5300 s to 644 s), supporting the hypothesis that UV-DDB facilitates glycosylase turnover from abasic sites. To delineate a more transient interaction between AAG and its target lesions, we turned to our novel SMADNE approach and utilized nick translation to create a lambda DNA substrate containing Hx at 10 different sites [32]. Briefly, lambda DNA was nicked 10 times by the nickase Nt.BspQI, and Pol I nick-translation was then used to incorporate dITP, the nucleotide form of Hx, and a fluorescent Cy3-labeled dUTP to use as a fiducial marker for the damaged DNA. Nuclear extracts were prepared from U2OS cells that had been transiently transfected with a plasmid-encoding GFP-tagged AAG. The GFP-tagged AAG was visualized at the single-molecule level on the LUMICKS C-trap with a green 488 laser, and kymographs were collected to study the interactions of GFP-AAG with Hx-containing DNA. We observed about 23% of the binding events to co-localize with Hx damage and calculated a binding lifetime of 2.8 s. To examine whether DDB2 and AAG formed co-complexes in cells during the repair of damage induced by methyl methane sulfonate (MMS), IF demonstrated discrete DDB2 and AAG foci formation, which peaked three hours post damage (Figure 4). Incomplete repair of alkylation damage is detrimental to cell survival. Using siRNA to transiently knock down either DDB2 or AAG in U2OS cells, we demonstrated decreased sensitivity to MMS treatment over an eight-day survival study [32]. These data suggest that repair intermediates generated during the repair of MMS damage are more toxic to cells than the initial alkylation damage [64].

## 4. Role of UV-DDB in 5-Hydroxymethyl-2-Deoxyuridine Repair

All the DNA bases are subjected to oxidative damage; one product of thymidine oxidation is the lesion 5-hydroxymethyl-deoxyuridine (5-hmdU). Single-strand selective monofunctional glycosylase (SMUG1) works to remove 5-hmdU, uracil, 5-fluorouracil, and oxidized thymidine derivatives from DNA. The name is a misnomer, as SMUG1 can recognize and process substrates on both single-stranded and double-stranded DNA with almost equal activities [65]. The enzymatic activity of SMUG1 on nucleosomes has been previously studied by the Delaney group, and SMUG1 activity on lesions within nucleosomes decreases three- to nine-fold [17]. Given the role of UV-DDB in promoting nucleosome destabilization after UV damage and chromatin decompaction after 8-oxoG, we wanted to elucidate the role of UV-DDB in SMUG1-mediated BER [29]. Sunbok Jang from our group used EMSAs to establish that UV-DDB is capable of specifically binding SMUG1 target substrates, dU and 5-hmdU, paired with deoxyadenosine or deoxyguanosine. Using 1.5 mM Mg^2+^ to improve the specificity of UV-DDB binding to these substrates, he observed a five- to six-fold increase in affinity compared with undamaged DNA (Figure 1). He then performed excision assays with SMUG1 on these substrates and demonstrated four- to five-fold stimulation of SMUG1 enzymatic activity with UV-DDB present (Figure 2). The next step was to define the mechanism of stimulation through biochemistry and single-molecule approaches [29]. Using native PAGE, we asked if UV-DDB can displace SMUG1 from the abasic site. Purified SMUG1 was incubated with THF-containing DNA as well as increasing amounts of UV-DDB, and we observed UV-DDB on the abasic site after promoting SMUG1 turnover. Unlike the other glycosylases stimulated by UV-DDB, we did not observe any co-complex formation between SMUG1 and UV-DDB using EMSAs. We then turned to single-molecule studies, specifically the DNA tightrope assay, to understand the transient interactions between SMUG1 and UV-DDB. Using differentially Qdot-labeled SMUG1 and UV-DDB, Jang was able to study their interactions on abasic sites. The presence of UV-DDB decreased the retention time of SMUG1 on abasic sites from 4950 s to 580 s. Our native PAGE experiments implied that the interactions between UV-DDB and SMUG1 are short-lived, and this was supported by the DNA tightrope assay. While we were able to demonstrate co-localization on abasic sites between UV-DDB and SMUG1, these interactions were rare, as they only occurred around 15% of the time. In cells, 5-hmdU can be generated via the direct oxidation of thymidine by the ten-eleven translocation enzymes [66]. Additionally, 5-hmdU is a nucleoside analog that can get phosphorylated and subsequently incorporated during DNA replication. Sripriya Raja, a Ph.D. student in our group, overexpressed SMUG-GFP and DDB2-mCherry constructs in U2OS cells and looked at their recruitment after 5-hmdU treatment via immunofluorescence and proximity ligation assays. After 5-hmdU treatment, we observed a 4.5-fold and 11-fold increase in SMUG1-GFP and DDB2-mCherry foci, respectively, via IF (Figure 4) [29]. To further validate that the interactions between SMUG1 and DDB2 are transient, we performed PLA to distinguish interactions between proteins within 40 nm of each other and demonstrated an increase in PLA signal after damage. Immunofluorescence was used to understand the kinetics of recruitment of SMUG1 and DDB2 after 5-hmdU damage. Consistent with DDB2 acting as a damage sensor, she observed early recruitment of DDB2 immediately after damage, while SMUG1 recruitment only peaked 30 min post damage. Interestingly, she saw DDB2 recruitment occurring in two waves, one which peaked 15 min post damage, and the second wave peaking two hours post damage. The later surge of DDB2 recruitment could be implications of UV-DDB having roles in downstream BER. As mentioned for alkylation damage, repair intermediates can sometimes be more detrimental to the cell than the original lesion. Using both short- and long-term growth assays, we demonstrated that cells deficient in SMUG1 or SMUG1/DDB2 are desensitized to 5-hmdU treatment. We observed sensitivity to 5-hmdU in DDB2 deficient cells, supporting the idea that DDB2 is required to facilitate the turnover of SMUG1 from abasic sites to promote efficient repair [29]. Long-term treatment of 5-hmdU resulted in striking toxicity in U2OS cells, which required further study. Key intermediates formed during BER include the abasic site created after the lesion is removed and the single-strand break created after abasic site processing. Single strand breaks in BER are bound by PARP1, leading to PARP1 activation and subsequent PARylation (PAR). The PAR chains are rapidly catabolized by the glycohydrolase, PARG [41]. We wanted to validate that our 5-hmdU treatment was initiating BER by measuring PAR accumulation. Using immunofluorescence, we saw a significant increase in PAR foci after treatment with 5-hmdU and a PARG inhibitor (PARGi) in WT cells, which was diminished in SMUG1-, DDB2-, and SMUG1/DDB2-deficient cells [29]. These data support the hypothesis that active repair of 5-hmdU is toxic to cells [67]. Lastly, since extensive PARylation can deplete cellular NAD, we wanted to assess the bioenergetic state of cells during active 5-hmdU repair. Previous studies have demonstrated that PARP1 activation during certain DNA damage responses can decrease oxidative phosphorylation [68]. Using the Seahorse Flux Analyzer, we measured basal oxygen consumption rates (OCR) and basal extracellular acidification rates (ECAR) after 5-hmdU and PARGi treatment in U2OS WT, SMUG1, DDB2, and SMUG1/DDB2 knockdown cells [29]. After 5-hmdU and PARGi treatment, we saw a significant decrease in the basal OCR, which were rescued by the addition of the PARP1 inhibitor (PARPi), Olaparib. Moreover, the loss of SMUG1 prevented the reduction of OCR after 5-hmdU and PARGi treatment. Interestingly the loss of DDB2 alone decreased basal OCR in untreated cells [29]. Taken together, these data support the hypothesis that SMUG1-DDB2-initiated repair of 5-hmdU can produce toxic intermediates, including PARP1 hyperactivation, leading to a reduction in oxidative phosphorylation.

## 5. Conclusions and Outlook

For the faithful duplication of our genome during cell division, single base damage must be efficiently repaired. Monofunctional glycosylases, such as MUTYH, AAG/MPG, and SMUG1, and bifunctional glycosylases with weak AP lyase activities, such as OGG1, require the help of other proteins to facilitate turnover and progress repair. These glycosylases also do not efficiently process damage when it is embedded in a nucleosome [9,13,15,16,17,18,23]. Over the last few years, our group has worked diligently to establish a role for UV-DDB in the base excision repair initiated by these glycosylases [28,29,30,31,32]. The biochemistry experiments established that UV-DDB can bind the target lesions of OGG1, MUTYH, AAG/MPG, and SMUG1 with high specificity and subsequently stimulate the enzymatic activities of the glycosylases. We used single-molecule approaches to conclude that the stimulation of these glycosylases by UV-DDB occurs through a facilitated dissociation mechanism, driven by competition for binding to the abasic site. Moreover, the single-molecule experiments revealed that UV-DDB can transiently interact with OGG1, MUTYH, AAG/MPG, and SMUG1 at abasic sites. After 8-oxoG, MMS, and 5-hmdU damage, we were able to see discrete recruitment of DDB2 with OGG1, AAG/MPG, and SMUG1, respectively, in cells. We observed the recruitment of other NER proteins, XPC and XPA, after 8-oxoG damage, with XPA recruitment being DDB2-independent but transcription-dependent [31]. DDB2 was shown to be necessary and sufficient to facilitate chromatin decompaction after 8-oxoG induction at the telomeres. Taken together, these data allowed us to establish that UV-DDB has a direct role in BER (Figure 5) as a damage-sensor and facilitates chromatin decompaction to increase lesion accessibility.

The studies covered in this review provide key molecular insights into how UV-DDB contributes to the BER initiated by OGG1, MUTYH, AAG/MPG, and SMUG1. However, future studies are necessary to understand how UV-DDB modulates chromatin structure changes to improve lesion accessibility and subsequent repair. While we demonstrated DDB2 having a role in chromatin decompaction at the telomeres after 8-oxoG damage, it would be interesting to determine if this regulation occurs exclusively in heterochromatic regions. There needs to be more work to understand the roles of DDB2 in processing 8-oxoG at euchromatin. The recruitment of the TC-NER protein, XPA, was dependent upon transcription; therefore, it would be interesting to determine if other TC-NER proteins, CSA and CSB, are recruited in response to 8-oxoG damage in euchromatic regions. Our studies with UV-DDB and MUTYH defined a necessary role for UV-DDB in the removal of 8-oxoG:A. Therefore, future studies could focus on understanding further roles of UV-DDB interacting with other long-patch BER proteins such as PCNA or Polλ. We were able to demonstrate the ability of UV-DDB to stimulate AAG/MPG activity; however, more work needs to be done to understand if UV-DDB can promote glycosylase activity when the alkylation damage is embedded within a nucleosome. It will be important to study the role of post-translational modifications, including ubiquitination by the DDB2-Cullin4A-RBX E3 ubiquitin ligase complex in changing chromatin structure. One possible mechanism is the post-translational modifications driven by UV-DDB, such as ubiquitylation, PARylation, and SUMOylation, which facilitate the recruitment of chromatin remodelers and histone modifier proteins, such as HPF1. PARylation during the processing of base damage is driven by PARP1; therefore, it would be of interest to study the interplay between UV-DDB and PARP1 during oxidative damage repair. In the context of UV damage, PARP1 has been proposed to help initiate NER via the PARylation of DDB2 to prevent degradation by suppressing the auto-ubiquitination and subsequent recruitment of the chromatin remodeler ALC1 [47]. Recently, it was shown that ALC1 is necessary for efficient BER. Therefore, the interplay between UV-DDB in the context of oxidative damage should be investigated [69]. A proteomics study indicated that PARP1 can interact with both DDB1 and DDB2 at stressed replication forks [70]. We demonstrated an increase in PARP1 activity after SMUG1-initiated BER; however, it would be interesting to study PARP1 activation after 8-oxoG damage. We demonstrated a decrease in cellular bioenergetics in cells depleted of DDB2. This change in metabolism as a direct role of UV-DDB in the removal of spontaneous base damage could explain why the loss of DDB2 in mice leads to spontaneous tumor formation and premature death [71,72].

Additionally, glycosylases such as NTH1 and NEIL1, which are not product-inhibited, can, however, become impaired if the target lesions are in inaccessible positions on nucleosomes [73,74]. Thus, future studies are necessary to understand the role of UV-DDB in the removal of other DNA lesions, such as thymine glycol, in chromatin. Our recently established SMADNE approach will be pivotal in answering fundamental questions about the dynamics of DDB2 on nucleosomes at the single-molecule level. The Lumicks C-trap provides the ability to study nucleosomal DNA at various tensions and rotational positions [75], allowing for a comprehensive understanding of the molecular dynamics and interactions during the processing of damage sites, and will help to define the precise role of ALC1 and other chromatin remodeling protein complexes in the efficient processing of base damage embedded in nucleosomes. Together, our recent studies have established a direct and early role for UV-DDB in the base excision repair of multiple types of base damage, working to increase lesion accessibility and promote stimulation of six different repair enzymes: OGG1, APE1, Pol β, MUTYH, AAG/MPG, and SMUG1.

## Figures and Tables

**Figure 1 ijms-24-10168-f001:**
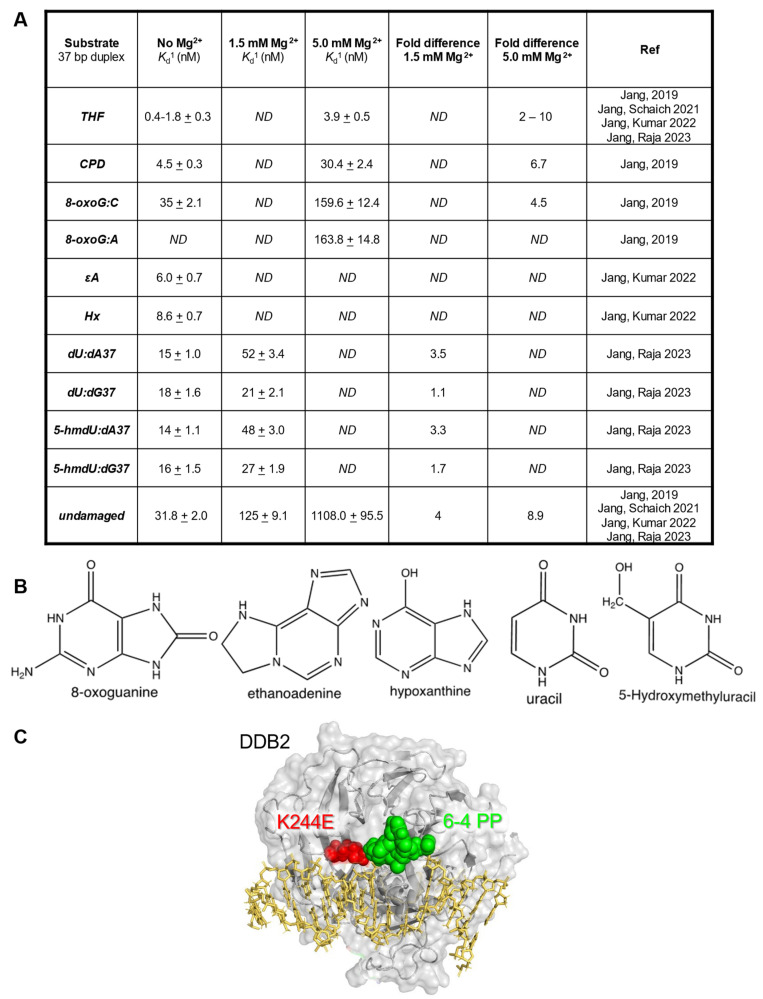
UV-DDB can bind BER substrates with high specificity. (**A**). Table of equilibrium dissociation constants (K_d_) for UV-DDB and various DNA damage substrates in the absence (0 mM) or presence (1.5 mM or 5 mM) of magnesium from [28,29,30,32]. (**B**). Structures of 8-oxoguanine, hypoxanthine, ethenoadenine, uracil, and 5-hydroxymethyluracil. (**C**). Structure of the xeroderma pigmentosum DDB2 variant, highlighting the K244E mutation, reprinted with permission from [51].

**Figure 2 ijms-24-10168-f002:**
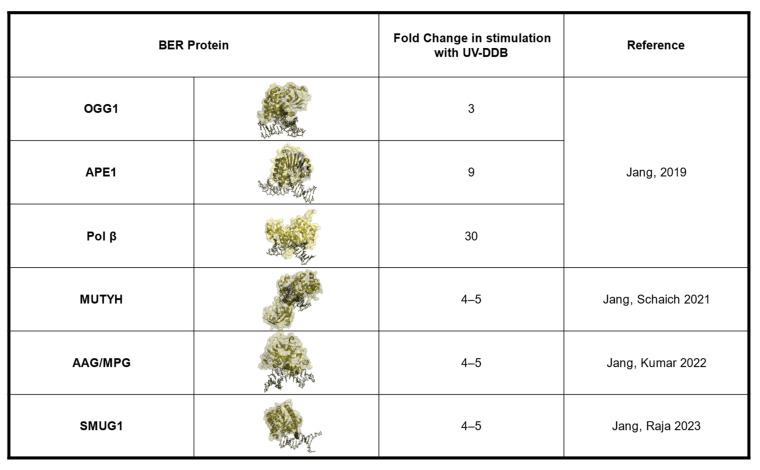
UV-DDB stimulates the activities of OGG1, MUTYH, AAG/MPG, and SMUG1. Fold change in enzymatic activity of OGG1, APE1, Pol β, MUTYH, AAG/MPG, and SMUG1 in the presence of UV-DDB, from [28,29,30,32].

**Figure 3 ijms-24-10168-f003:**
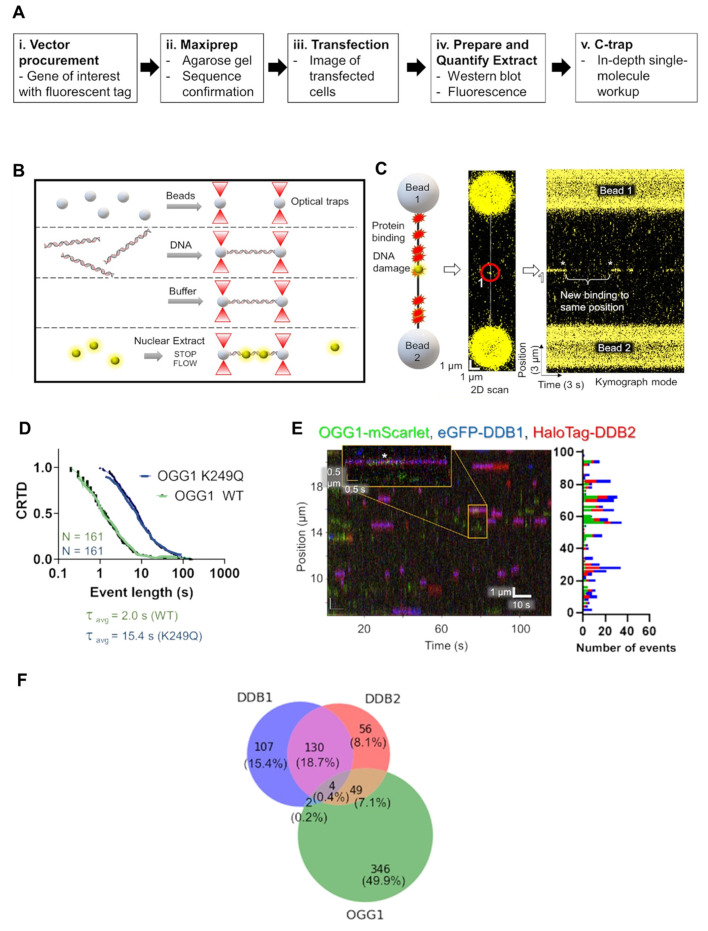
SMADNE approach to show UV-DDB and OGG1 interactions on C-trap. (**A**). SMADNE workflow. (**B**). Experimental setup for C-trap optical tweezer system. (**C**). Cartoon rendering of the streptavidin-coated beads used to tether DNA substrate of interest. (**D**). Cumulative residence time distribution (CRTD) analysis of WT and catalytically dead OGG1 on 8-oxoG containing DNA. (**E**). Kymograph of OGG1-mScarlet (green), eGFP-DDB1 (blue), and HaloTag-DDB2 (red) with binding positions (right). The white asterisk (*) is marking co-localization between OGG1, DDB1, and DDB2. (**F**). Venn diagram showing the distribution of events, individual versus colocalization. Adapted with permission from [51].

**Figure 4 ijms-24-10168-f004:**
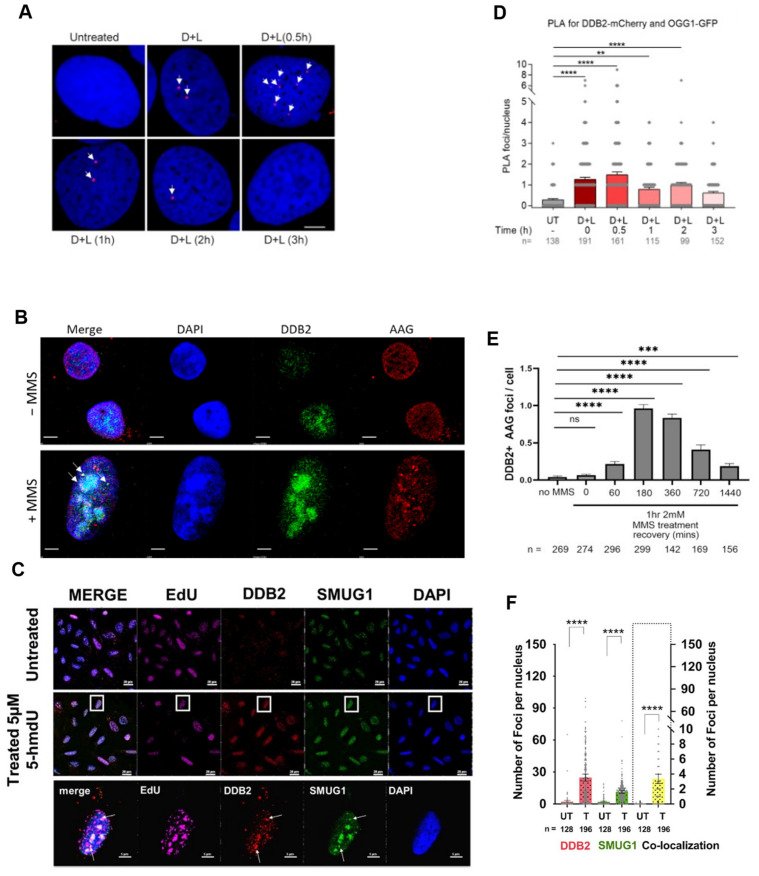
UV-DDB co-localizes with OGG1, AAG, and SMUG1 during BER in cells. Representative images showing co-localization of DDB2, the DNA binding subunit of UV-DDB, with (**A**). OGG1, proximity ligation assay (PLA) between DDB2-mCherry and OGG1-GFP, One-way ANOVA (Sidak multiple comparison test): ** *p*  <  0.01, **** *p*  <  0.0001). (**B**). AAG, Immunofluorescence (IF) IF of mNeonGreen-tagged DDB2 and endogenous AAG (ns: *p* > 0.05, ***: *p* < 0.001, ****: *p* < 0.0001, *t*-test), and (**C**). SMUG1, IF staining for mCherry and GFP, and Alexa 647 staining of EdU (**** *p* < 0.0001; unpaired *t*-test) after specific oxidative damage, and the respective quantitation (**D**–**F**). “n” represents the number of cells scored for each condition; arrows identify points of co-localization between DDB2 and the glycosylase. Scale bar is 5 µm for all images. Adapted with permission from: [29,31,32].

**Figure 5 ijms-24-10168-f005:**
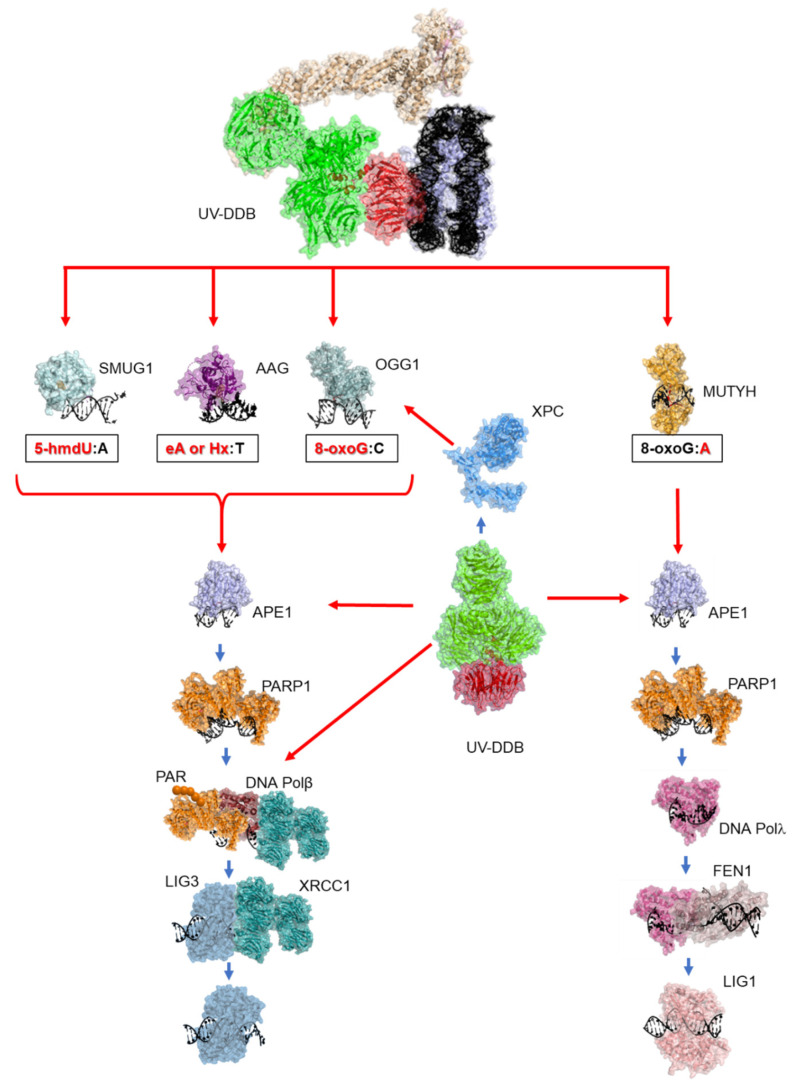
Working model of UV-DDB in BER in the context of chromatin. In response to single base damage, UV-DDB can act as a damage sensor, and can work to stimulate the activities of OGG1, MUTYH, AAG/MPG, and SMUG1 glycosylases (perhaps through XPC), APE1, and Pol β (red arrows) to facilitate BER (blue arrows).

## Data Availability

Not applicable.

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
