# Peer review of "UV-DDB as a General Sensor of DNA Damage in Chromatin: Multifaceted Approaches to Assess Its Direct Role in Base Excision Repair"

_ijms, 2023, doi:10.3390/ijms241210168_

Round 1

Reviewer 1 Report

This comprehensive review provides an insightful overview of the studies performed in the recent years by Bennett Van Houten and colleagues to explore the role of UV-DDB2 in the repair of single base lesions by BER DNA glycosylases. The review is well written and structured, and provides a suitable level of detail. The text is supported by high quality illustrations.

Authors should ensure that all appropriate references are included. References are missing for example in the first paragraph of section 3, and when describing their work on the decompaction effect of DDB2 in telomeres at the end of section 2.

Line 259: typo fuorgen

Title of section 4, line 410: Uv-Ddb should be in capital letters to be consistent with the rest of the manuscript

Suggestion: combine sections 3 and 4 and start with oxidized thymines as a more logical follow-up of 8-oxo-guanine, before moving to alkylated bases.

Line 411-412: the oxidation of thymidine does not only create 5-hydroxymethyl-deoxyuridine. This should be clarified or rephrased to avoid any confusion.

Author Response

This comprehensive review provides an insightful overview of the studies performed in the recent years by Bennett Van Houten and colleagues to explore the role of UV-DDB2 in the repair of single base lesions by BER DNA glycosylases. The review is well written and structured and provides a suitable level of detail. The text is supported by high quality illustrations.

We appreciate this reviewer’s enthusiasm and kind feedback for the manuscript.

Authors should ensure that all appropriate references are included. References are missing for example in the first paragraph of section 3, and when describing their work on the decompaction effect of DDB2 in telomeres at the end of section 2.

We thank the reviewer for this feedback and have added the appropriate references as requested  

Line 259: typo fluorgen

Title of section 4, line 410: Uv-Ddb should be in capital letters to be consistent with the rest of the manuscript

We thank the reviewer for finding these typos and have methodically studied the manuscript for any additional typos.

Suggestion: combine sections 3 and 4 and start with oxidized thymines as a more logical follow-up of 8-oxo-guanine, before moving to alkylated bases.

We appreciate this suggestion, however we chose the order and distribution of the sections to correspond with the historical publication order of the manuscript.

Line 411-412: the oxidation of thymidine does not only create 5-hydroxymethyl-deoxyuridine. This should be clarified or rephrased to avoid any confusion.

We thank the reviewer for pointing this out and have rewritten this sentence for clarity.  

Reviewer 2 Report

In the manuscript “UV-DDB as a general sensor of DNA damage in chromatin: a direct role in base excision repair” the authors present different approaches carried out in their lab to investigate the role of UV-DDB, a protein already involved in NER, in BER and in particular in regulating the turnover of different glycosylases involved in BER in a chromatin context. 

The review briefly describes the role of glycosylases in BER and the effects of nucleosomes in the recognition of damaged nucleotides by these enzymes, then focusing on the mechanisms that allow chromatin decompaction and DNA damage recognition. The authors explain in detail the different approaches that people in their lab set up to unveil the role of UV-DDB in these processes and the results they obtained in the last few years. The experimental approaches are well described, considering the purpose of the manuscript. I found this review is an important tool for researchers who study DNA repair mechanisms and the impact of chromatin structure in DNA damage recognition, although I found is a bit self-congratulatory. Therefore, I support its publication in IJMS. 

I have some minor points, listed below:

1-    I think the main focus of this review is the power of combination of multifaceted approaches to assess the role of UV-DDB in BER. I the caught this focus from the abstract, while in my opinion the title suggests a more general and theorical view. Could the authors consider modifying the title introducing the methodological view of the review? E.g. “UV-DDB as a general sensor of DNA damage in chromatin: a multifaceted approach to assess a direct role in base excision repair”?

2-    The manuscript is well organized. However, I found some sections are a bit long. Therefore, I suggest introducing the sub-sections where possible to help the readers. E.g., section 2 could be divided in subsections for the different approaches. 

3-    Lane 111: remove the dot between “nucleosome” and “and”

4-    Lane 410: Uv-Ddb in capital letters

Author Response

Reviewer 2:

In the manuscript “UV-DDB as a general sensor of DNA damage in chromatin: a direct role in base excision repair” the authors present different approaches carried out in their lab to investigate the role of UV-DDB, a protein already involved in NER, in BER and in particular in regulating the turnover of different glycosylases involved in BER in a chromatin context. 

The review briefly describes the role of glycosylases in BER and the effects of nucleosomes in the recognition of damaged nucleotides by these enzymes, then focusing on the mechanisms that allow chromatin decompaction and DNA damage recognition. The authors explain in detail the different approaches that people in their lab set up to unveil the role of UV-DDB in these processes and the results they obtained in the last few years. The experimental approaches are well described, considering the purpose of the manuscript. I found this review is an important tool for researchers who study DNA repair mechanisms and the impact of chromatin structure in DNA damage recognition, although I found is a bit self-congratulatory. Therefore, I support its publication in IJMS. 

We appreciate this reviewer’s enthusiasm and kind feedback for the manuscript.

I have some minor points, listed below:

1-    I think the main focus of this review is the power of combination of multifaceted approaches to assess the role of UV-DDB in BER. I the caught this focus from the abstract, while in my opinion the title suggests a more general and theorical view. Could the authors consider modifying the title introducing the methodological view of the review? E.g. “UV-DDB as a general sensor of DNA damage in chromatin: a multifaceted approach to assess a direct role in base excision repair”?

We thank the reviewer for this great suggestion, and we have updated the title.

2-    The manuscript is well organized. However, I found some sections are a bit long. Therefore, I suggest introducing the sub-sections where possible to help the readers. E.g., section 2 could be divided in subsections for the different approaches. 

3-    Lane 111: remove the dot between “nucleosome” and “and”

4-    Lane 410: Uv-Ddb in capital letters

We thank the reviewer for this suggestion and added subsection headings to section 2 and corrected the typos.